# A Conserved Tryptophan in the Ebola Virus Matrix Protein C-Terminal Domain Is Required for Efficient Virus-Like Particle Formation

**DOI:** 10.3390/pathogens9050402

**Published:** 2020-05-22

**Authors:** Kristen A. Johnson, Rudramani Pokhrel, Melissa R. Budicini, Bernard S. Gerstman, Prem P. Chapagain, Robert V. Stahelin

**Affiliations:** 1Department of Chemistry and Biochemistry, University of Notre Dame, Notre Dame, IN 46556, USA; Kristen.Johnson@UTSouthwestern.edu (K.A.J.); Melissa.Budicini@UTSouthwestern.edu (M.R.B.); 2Physics Department, Florida International University, Miami, FL 33199, USA; rpokh002@fiu.edu (R.P.); gerstman@fiu.edu (B.S.G.); chapagap@fiu.edu (P.P.C.); 3Biomolecules Sciences Institute, Florida International University, Miami, FL 33199, USA; 4Department of Medicinal Chemistry and Molecular Pharmacology and the Purdue Institute for Inflammation, Immunology and Infectious Disease, Purdue University, West Lafayette, IN 47907, USA

**Keywords:** Ebola virus, lipid–protein interaction, oligomerization, phosphatidylserine, phosphatidylinositol-4,5-bisphosphate, tryptophan, virus assembly, virus budding, VP40

## Abstract

The Ebola virus (EBOV) harbors seven genes, one of which is the matrix protein eVP40, a peripheral protein that is sufficient to induce the formation of virus-like particles from the host cell plasma membrane. eVP40 can form different structures to fulfil different functions during the viral life cycle, although the structural dynamics of eVP40 that warrant dimer, hexamer, and octamer formation are still poorly understood. eVP40 has two conserved Trp residues at positions 95 and 191. The role of Trp^95^ has been characterized in depth as it serves as an important residue in eVP40 oligomer formation. To gain insight into the functional role of Trp^191^ in eVP40, we prepared mutations of Trp^191^ (W191A or W191F) to determine the effects of mutation on eVP40 plasma membrane localization and budding as well as eVP40 oligomerization. These in vitro and cellular experiments were complemented by molecular dynamics simulations of the wild-type (WT) eVP40 structure versus that of W191A. Taken together, Trp is shown to be a critical amino acid at position 191 as mutation to Ala reduces the ability of VP40 to localize to the plasma membrane inner leaflet and form new virus-like particles. Further, mutation of Trp^191^ to Ala or Phe shifted the in vitro equilibrium to the octamer form by destabilizing Trp^191^ interactions with nearby residues. This study has shed new light on the importance of interdomain interactions in stability of the eVP40 structure and the critical nature of timing of eVP40 oligomerization for plasma membrane localization and viral budding.

## 1. Introduction

The Ebola virus (EBOV) was first discovered in 1976 following an outbreak of hemorrhagic fever near the Ebola River in Zaire [1]. Since that time there have been sporadic outbreaks in Africa including the largest EBOV outbreak in history in Western Africa (2014–2016) and a current outbreak in the Democratic Republic of Congo. EBOV causes significant disease pathologies such as hemorrhage and organ failure and often has a fatality rate >50% [2]. While there are not yet FDA or WHO approved drugs or antibodies for disease treatment, a vaccine was approved in December of 2019 [3]. 

EBOV is a filamentous lipid enveloped virus from the Filoviridae family. EBOV harbors a negative sense RNA genome that encodes seven genes [4] and the replication and formation of new virus particles from the infected host cell requires the EBOV matrix protein VP40 (eVP40) [5,6,7]. eVP40 is able to induce virus-like particle (VLP) formation from the plasma membrane of human cells in the absence of other filovirus proteins [7,8,9]. eVP40 is a peripheral protein that was initially shown to associate with the anionic lipid phosphatidylserine (PS) [10], where membrane binding was proposed to induce conformational changes in the eVP40 structure [11]. Indeed, more recent studies have indicated that PS is critical to the association of eVP40 to the plasma membrane inner leaflet where PS levels increased eVP40 oligomers [12,13] required for VLP formation. Once eVP40 has engaged PS, eVP40 oligomers form and interact with phosphatidylinositol-4,5-bisphosphate (PI(4,5)P_2_), which has been shown to stabilize eVP40 oligomers [14]. The interactions of eVP40 oligomers with PI(4,5)P_2_ are also critical to viral egress as disrupting PI(4,5)P_2_ levels in the plasma membrane significantly hinder VLP formation [14]. 

eVP40 is a transformer protein of 326 amino acids that has been shown to form different protein structures for different functions in the viral life cycle [15]. eVP40 is a dimer that contains an N-terminal domain (NTD) involved in dimerization and a C-terminal domain (CTD) that regulates membrane binding. Additionally, both the NTD and CTD participate in interdomain interactions for eVP40 hexamers [15,16,17] and larger oligomers [18,19,20,21] that form at the membrane interface. An eVP40 octameric ring structure has also been identified that is mediated by a different NTD oligomerization interface [15,21]. The eVP40 octameric ring binds RNA [15,21] and has been proposed to regulate the rate of viral transcription. eVP40 octamers also play a role distinct from assembly and budding as the octamers have not been detected at the plasma membrane or in viral particles. A number of studies have implicated different regions or amino acids of eVP40 to play critical roles in processes such as membrane binding [13,15,19,20], oligomerization [15,17] and protein–protein interactions [22,23,24]. While a number of key interactions between VP40 and host molecules have been found, much less is known regarding the structural dynamics that regulate eVP40 oligomeric states and elements of the eVP40 structure critical to protein stability or conformational change.

eVP40 harbors two tryptophan residues, one at position 95 (Trp^95^) and the second at position 191 (Trp^191^). Trp^95^ has been investigated in a number of studies and implicated in the formation of oligomers necessary for formation of new virus particles [15,18,25]. Trp^95^ lies at an NTD–NTD interface critical to formation of VP40 hexamers, which are required for viral budding from the plasma membrane of the host cell [15,18]. In contrast, the functional role (if any) of Trp^191^ is unknown. Trp^191^ is located in a linker region between the NTD and CTD of VP40 and protrudes toward a pocket just below important cationic residues, which are required for the binding to plasma membrane anionic lipids [13,15]. Trp^191^ may play a functional role in lipid binding or stabilizing the eVP40 structure for effective localization and interaction with membranes (Figure 1). However, the role or importance of Trp^191^ in eVP40 membrane binding, oligomerization, or matrix assembly is not known. To gain insight into the role of Trp^191^ in eVP40, we prepared mutations of Trp^191^ to Ala and Phe and examined eVP40 functional changes using cellular and in vitro biochemical and biophysical approaches, as well as computational studies. 

## 2. Results

### 2.1. Mutation of Trp^191^Alters Plasma Membrane Localization, VP40 Oligomerization, and Virus-Like Particle Formation

To investigate the role of Trp^191^ in eVP40, we prepared W191A and W191F mutations of eVP40 to monitor eVP40 plasma membrane localization, VLP formation, eVP40 oligomerization, and eVP40 membrane binding. Trp mutations to Ala and Phe were made to remove the aromatic and bulkiness of the Trp residue (Ala) while also retaining some elements of the aromatic character (Phe). We first compared the plasma membrane localization of wild-type (WT) EGFP–VP40 to that of EGFP–W191A and EGFP–W191F (Figure 2A). COS-7 cells, which have previously been used for different aspects of filovirus research [14,26,27,28,29] as non-human primates can also be infected by Ebola virus, were imaged 12–14 h post-transfection. The plasma membrane localization and pre-VLP formation of the three different constructs was then quantified as shown in Figure 2A,B. W191F and W191A both led to a statistically significant reduction in plasma membrane localization as well as evidence of pre-VLP formation (marked with white arrows in Figure 2A). Notably, W191A had some cells with a clear lack of pre-VLP formation (see Figure 2A inset) and most cells had the appearance of eVP40 positive perinuclear puncta. eVP40 puncta have previously been observed for mutations that stall or induce changes in eVP40 trafficking [30] as well as in cases where eVP40 octameric ring formation is favored [13,15]. While the W191F lacked appearance of detectable intracellular puncta, some cells expressing W191F lacked pre-VLP formation (see Figure 2A inset).

Next, we wanted to determine if either mutation at Trp^191^ altered the formation of VLPs (i.e., the scission step from the plasma membrane). As perhaps expected from the cellular imaging experiments, W191A led to a 70% reduction in VLP formation compared to WT VP40 whereas W191F reduced VLP formation but not to a statistically significant level (Figure 2C,D). This suggests that Trp at position 191 is an important contributor to the proper plasma membrane localization of VP40 and VLP formation. Phenylalanine, likely due to its aromatic nature, is partially able to substitute for tryptophan at this position, as despite a significant reduction in plasma membrane localization, VLPs can still be formed at near WT levels for this mutation.

eVP40 oligomerization is a hallmark of its ability to form pre-VLP structures that emanate from the plasma membrane [12,14,15,18]. Proper oligomerization of eVP40 from the dimer to a hexamer [12,15] and larger oligomers [12,14,18,19,20] has been proposed to be a key step in proper egress and virion formation, as mutations that reduce oligomerization or alter hydrophobic contacts that facilitate proper CTD oligomer contacts reduce budding [15,16,17]. To monitor the effects of W191A and W191F on VP40 oligomeric state, we employed number and brightness (N&B) analysis, which has been used to study the apparent oligomerization state of GFP tagged proteins [31,32,33], including previous work on VP40 [12,14,18,19,20]. COS-7 cells expressing WT, W191A, or W191F as EGFP fusions were imaged using a raster scan for 100 frames (Figure 3A) and the resulting images were analyzed using Globals software (SimFCS, E. Gratton, UC Irvine Laboratory for Fluorescence Dynamics, Irvine, CA, USA) and the N&B analysis package. In line with previous studies [12,14,18,19,20], WT VP40 exhibited significant oligomerization at sites of pre-VLP formation (Figure 3A,C). WT VP40 exhibited significant populations of hexamer–12mers (green), and 12mers and greater (blue) at pre-VLP sites (Figure 3A,C). To visualize the differences more simply between WT and mutant oligomerization, WT VP40 oligomerization was normalized to 100% for all three oligomer sizes monitored to detect changes in W191A or W191F oligomerization. 

W191A, which showed low pre-VLP formation in the image stack shown in Figure 3A, exhibited a slight reduction in monomer–hexamer formation and a statistically significant increase in hexamer–12mer formation (Figure 3B–D). The 12mer and greater population was similar overall to wild-type but more variable as evidenced by the larger error bars in Figure 3D. Thus, we hypothesized that since W191A had a low level of plasma membrane localization and VLP formation, the W191A mutation increased the VP40 octamer populations consistent with earlier observed cellular punctate with this mutation. The W191F mutation in contrast behaved more like WT with only a small increase in the hexamer–12mer population and a slight decrease in the 12mer and greater population (Figure 3D). The lower level of 12mer and greater population of W191F, although not statistically significant, is consistent with the decreased levels of pre-VLP and VLP formation observed earlier for this mutation. 

To gain further insight into how Trp^191^ mutations affect the VP40 oligomerization state, we expressed and purified VP40 and Trp^191^ mutations from *Escherichia coli* using affinity chromatography [13,14]. We then subjected all proteins to size exclusion chromatography as VP40 has previously been shown to purify as dimer and octamer [14,15]. As shown in Figure 4A, WT VP40 was purified as a dimer and octamer. In contrast, very little dimer was detected for W191A or W191F where both mutations led to a predominant in vitro population of octamer (Figure 4A). Thus, both mutations seemed to loosen or destabilize the NTD–CTD contacts that stabilize the dimer and likely repelled the CTD in such a way that the NTD was more accessible to octamer formation. This is somewhat in contrast to cellular data where the proposed octamer was significantly accumulated for W191A but at a lower level than for that of W191F. However, the in vitro system lacks components such as membrane binding or post-translational modifications that may stabilize VP40 dimers, hexamers, or other VP40 structures detected in cellular imaging. Nonetheless, mutation of Trp^191^ is sufficient to destabilize VP40 in the apo structure and clearly increased VP40 octamer formation relative to that of the dimer.

### 2.2. Mutation of Trp^191^ Significantly Reduces VP40 Lipid Binding Ability

The Trp^191^ residue is just below several previously identified Lys residues that were critical to VP40 anionic lipid binding and plasma membrane localization [13,15]. W191A and W191F reduced plasma membrane localization and VLP formation to different extents, which could be partially dependent upon binding to anionic lipids in the plasma membrane. To determine if mutations of Trp^191^ altered VP40 membrane binding, we employed a well-established lipid vesicle centrifugation assay [14] to determine if the mutations altered VP40 binding to PI(4,5)P_2_. PI(4,5)P_2_ has been shown to be an important component of VP40 VLP formation and VP40–lipid interactions at the plasma membrane inner leaflet [14]. As shown in Figure 4B,C, WT VP40 bound significantly to PI(4,5)P_2_ containing vesicles where approximately 50% of the protein was found in the lipid-bound pellet fraction. Similarly, W191F behaved like WT exhibiting comparable levels of protein in the lipid-bound fraction (Figure 4B,C). W191F also had a decrease in the amount of protein bound to control vesicles, suggesting the Trp residue may play some role in zwitterionic lipid binding, consistent with the role of Trp in previous studies on lipid-binding proteins [34,35]. W191A bound akin to both control and PI(4,5)P_2_ vesicles (Figure 4B,C) suggesting an aromatic residue at this position is essential to proper membrane binding. Thus, even though W191A and W191F were predominantly octameric, they had distinct lipid binding properties with W191F binding similar to WT and W191A lacking detectable binding to PI(4,5)P_2_ containing vesicles. 

### 2.3. Molecular Dynamics Simulations of WT VP40 and W191A

Molecular dynamics (MD) simulations have been used to investigate VP40 structural dynamics [16,17], VP40–lipid interactions [36,37], and to better understand cellular and experimental observations of VP40 structure/function relationships [13,38,39]. To determine how Trp^191^ may influence VP40 flexibility and interactions adjacent to Trp^191^, we performed MD simulations for WT VP40 and W191A. For the WT, residues in what we refer to as loop 1, containing key lipid-binding residues Lys^224^ and Lys^225^, are quite flexible (Figure 5A and Appendix A) whereas loop 2 residues (Lys^274^ and Lys^275^) are more restricted in their movements. The linker region between the NTD and CTD is also quite rigid and lacks flexibility in the WT simulations. However, simulations with the W191A mutant exhibited a K221–D230 salt-bridge form that staples loop 1 in a much more restrained conformation (Figure 5A, Figure 6, and Appendix A), significantly affecting the flexibility of the loop 1 residues. In contrast, loop 2 flexibility does not appear to change considerably in W191A but does have a slight change in orientation. The linker region between the NTD and CTD is much more flexible (Figure 5, Figure 7, and Appendix A) in W191A, which may partially explain the greater propensity of octamer formation for the W191A construct. The increased flexibility may destabilize the interdomain interactions and affect the CTD disengagement from the NTD. Diminished membrane binding and enhanced CTD disengagement would be expected to increase VP40 octamer ring formation, plasma membrane localization, and reduced VLP formation.

## 3. Discussion

The EBOV matrix protein has been the subject of much study as it is required for formation of the viral lipid-envelope from the host cell plasma membrane. To date, structure function studies of eVP40 have shed light on lipid binding residues required for plasma membrane binding or localization as well as VLP formation [13,15,19,20,40]. Additional studies have identified the origins of protein–protein interactions [22,23,24,41] or mutations that alter eVP40 trafficking [30,42]. Despite detailed investigations aimed at how the different eVP40 structures that form interact with host components or perform their cellular function, little data is available on the dynamics of different eVP40 structures, shifts in their equilibria, and residues or regions in eVP40 that may regulate eVP40 conformational dynamics and the different eVP40 structures that form. 

eVP40 is susceptible to being destabilized to an oligomeric form as early work in the field identified that increasing concentrations of urea increased eVP40 oligomer formation [11]. eVP40 also consistently purifies as a dimer and octamer when including RNAse treatment as shown in Figure 4 as well as published previously [13,15]. Additionally, mutations of the NTD dimer interface have been shown to increase not only the eVP40 monomer population but that of the octamer as well [13,15]. Lastly, a mutation of Ile^307^ in the eVP40 CTD leads predominantly to octamer formation [13,15]. Notably, structure and function studies of eVP40 have led to mutations that selectively allow study of one eVP40 population over another (e.g., dimer vs. octamer); however, the biophysics of eVP40 conformational changes and internal interactions that contribute to the stability of different eVP40 structural states is poorly understood. From these studies, it is clear that Trp^191^, and likely the composition of the NTD-CTD linker, are critical components of dimer stability. 

As shown in Figure 5 and Appendix A, Trp^191^ clearly impacts the flexibility and positioning of loops 1 and 2, both of which are required for plasma membrane binding and proper budding of VLPs [13,15]. The positioning of Trp^191^ may also be critical post-lipid binding, as lipid binding has been suggested to trigger eVP40 hexamer and larger assemblies [15,18,19,20] that are required for VLP formation. From our current model, it is possible that lipid binding to loops 1 and 2 above Trp^191^ and the NTD–CTD linker shown in Figure 5, repositions the NTD–CTD linker to favor separation of the NTD and CTD to help facilitate the assembly of larger eVP40 oligomers. 

Although it does not seem likely that Trp^191^ is directly involved in lipid binding, this cannot be discounted as Trp has a propensity to interact with zwitterionic lipids [34,35]; aromatic amino acids often compose parts of phosphoinositide-binding pockets [43] and can provide structural stability or contribute energy to the lipid binding interactions. Further, W191A led to loss of lipid binding while for the most part, lipid binding by W191F was retained. The molecular dynamics studies herein support the in vitro and cellular work suggesting the main role of Trp^191^ is to stabilize local hydrophobic interactions, which provides flexibility for loop 1 to interact with lipids. Disruption of local Trp^191^ interactions clearly induced loss of dimer stability in in vitro studies and increased the appearance of the octamer. 

Overall, the studies on Trp^191^ reported here help to better understand the importance of interdomain interactions in eVP40 stability and the delicate balance of internal interactions in stabilizing the eVP40 dimer structure for proper trafficking and function in cells. These studies also inform the importance of interdomain interactions, specifically those stabilized by Trp to stabilize protein structures that may regulate conformational changes or interactions with other ligands such as lipids. Future studies geared at mapping the phosphorylation [44], acetylation [45], and ubiquitylation [41,46] of eVP40 should enlighten our understanding of the structure and function relationship of eVP40 cellular oligomerization and mechanisms by which post-translational modifications favor different eVP40 structural forms. 

## 4. Materials and Methods

### 4.1. Molecular Biology

Site directed mutagenesis of VP40 in the pET-46 and (EGFP)-pcDNA3.1 expression vectors were achieved with the Agilent Quick Change XL kit (Agilent Technologies, Santa Clara, CA, USA). The following primers were used to generate W191F (forward) 5′-CCA CTG CCT GCT GCA ACA TTC ACC GAT GAC ACT CCA ACA GG-3′ (reverse) 5′-CCT GTT GGA GTG TCA TCG GTG AAT GTT GCA GCA GGC AGT GG-3′. The following primers were used to generate W191A (forward) 5′-CCA CTG CCT GCT GCA ACA GCG ACC GAT GAC ACT CCA ACA GG-3′ (reverse) 5′-CCT GTT GGA GTG TCA TCG GTC GCT GTT GCA GCA GGC AGT GG-3′. Sanger sequencing was used to confirm successful mutants (Notre Dame Genomics Core).

### 4.2. Cell Culture

COS-7 cells were maintained in DMEM with 10% fetal bovine serum and 1% Pen/Strep in an incubator with 5% CO_2_ at 37 °C. For imaging experiments, cells were seeded in 8 well Lab-Tek II Chambered Coverglass sterile imaging plates (Thermo Fisher, Waltham, MA, USA). Once cells reached 70–90% confluency they were transfected with 0.4 µg endotoxin free DNA and Lipofectamine 2000 (Life Technologies, Carlsbad, CA, USA) for 12–14 h. For VLP collections, cells were seeded in 100 mm dishes and transfected once 70–90% confluent with 19 µg DNA for 22–26 h. VLPs and cells were collected as described previously in detail [14]. 

### 4.3. Western Blot

VP40 budding efficiency was determined with Western blot as described previously in detail [16] with the exception of the primary antibody used to detect EGFP–VP40: mouse anti-EGFP, F56-6A1.2.3, (Thermo Fisher Scientific, Waltham, MA, USA) antibody and sheep anti-mouse HRP (AB 6808, Abcam, Cambridge, United Kingdom) secondary were used. For loading purposes, 100 ng of total protein was loaded onto the gels for Western blot as determined by a BCA assay.

### 4.4. Confocal Imaging

For cell counting and phenotype imaging, a Zeiss LSM 710 was used (IUSM Imaging Facility). For N&B imaging, an Olympus FV1000 was used as described previously [14,47]. N&B analysis was performed in SimFCS. Three experiments were performed for each imaging experiment. 

### 4.5. Protein Expression and Purification

VP40 protein was expressed and purified as described [14]. Eluted protein was further purified using size exclusion chromatography (GE Hi-Load S200 column) and the VP40–octamer fraction was collected and concentrated. A standard BCA protein assay kit (Thermo Fisher) was used to determine protein concentration. Protein was used within two weeks of purification to ensure full activity. 

### 4.6. Lipid Binding

All lipids were purchased from Avanti Polar Lipids (Alabaster, AL, USA). A liposome pelleting assay was used to determine VP40 binding to PI(4,5)P_2_. Lipids were measured into vials to achieve desired lipid compositions (control: 50% POPC, 48% DOPE, and 2% dansyl PE; PI(4,5)P_2_: 47.5% POPC, 45.5% DOPE, 5% PI(4,5)P_2_, and 2% dansyl PE). Lipids were dried under N_2_ gas and stored at −20 °C until use. The lipids were hydrated with a 250 mm raffinose pentahydrate solution (150 mM NaCl, 10 mM Tris, pH 7.4) then extruded through a 200 nm filter with an Avanti mini extruder (Avanti Polar Lipids). After size verification with dynamic light scattering, liposomes were diluted in raffinose free buffer for a final concentration of 0.6 mM lipid (150 mM NaCl, 10 mM Tris, pH 7.4) and incubated with 0.6 µM protein for 30 min at room temperature. Liposomes were pelleted with centrifugation (30 min, 75,000× *g*, 22 °C). Pellets and supernatants were assessed for VP40 using SDS PAGE. 

### 4.7. Molecular Dynamics Simulations

The X-ray crystal structure in Protein Data Bank (PDB ID 4LDB) was used for the structure of the wild-type (WT) eVP40 dimer (Chain A and Chain C). Modeller [48] was used to insert the missing residues. For the molecular dynamics (MD) simulations, we set up two systems, the wild-type (WT) dimer and the mutant dimer (W191A). Both systems were solvated and ionized using Charmm-Gui [49,50]. Each system was minimized for 10,000 steps followed by 200 ps of NPT equilibration and were simulated for 100 ns of production run. All-atom MD simulations were performed using NAMD2.12 [51] with the CHARMM36 force field [52] and 2 fs time step. Periodic boundary conditions were employed in all simulations. The long-range electrostatic interactions were treated using the particle mesh Ewald (PME) method [53] and the SHAKE algorithm was used to constrain the covalent bonds involving hydrogen atoms. A Nose–Hoover Langevin-piston method was used with a piston period of 50 fs and a decay of 25 fs to control the pressure. The temperature was controlled using Langevin temperature coupling with a friction coefficient of 1 ps^−1^. Visual molecular dynamics (VMD) [54] was used to analyze the trajectories.

## Figures and Tables

**Figure 1 pathogens-09-00402-f001:**
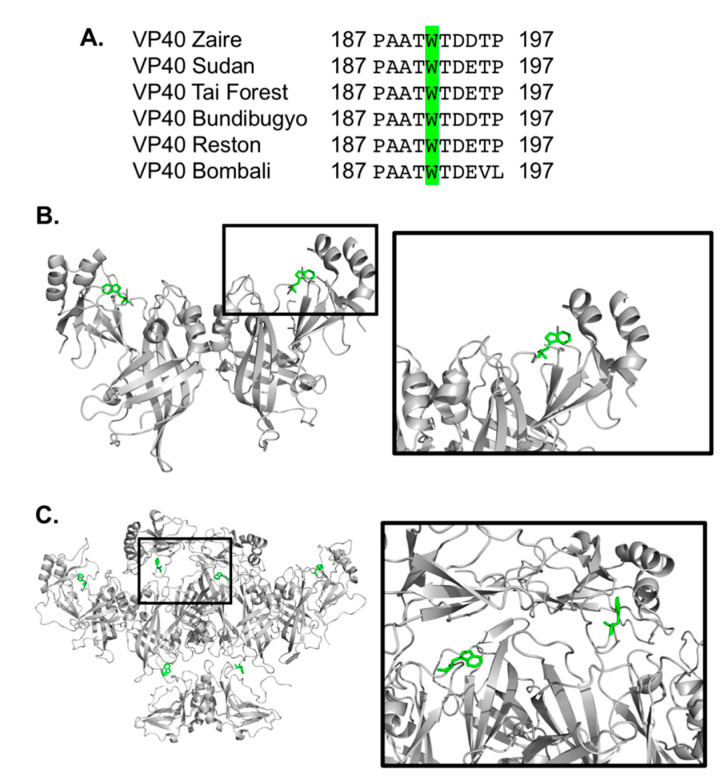
eVP40 has a conserved tryptophan at position 191. (**A**) Clustal W was used to align the amino acid sequences of eVP40 from six different types of ebolavirus. The alignment between residues 187 and 197 is shown with the conserved tryptophan residues (Trp^191^) highlighted in green. (**B**) The eVP40 dimer structure (PDB ID: 4LDB) was used to highlight the position of Trp^191^, which is situated near a C-terminal domain (CTD) region beneath Lys^274^ and Lys^275^, which are important for eVP40 plasma membrane binding [13,15]. (**C**) The VP40 hexamer structure (PDB ID: 4LDD) was used to model in the missing CTDs (Modeller software) to highlight the position of Trp^191^ in the proposed hexamer structure. Figure 1B,C were prepared in Pymol.

**Figure 2 pathogens-09-00402-f002:**
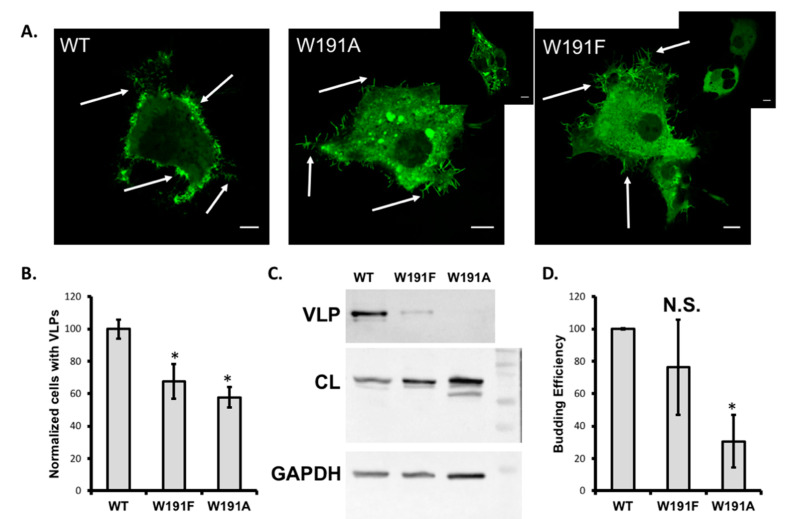
Plasma membrane localization and virus-like particle (VLP) formation of wild-type (WT) and VP40 eTrp^191^ mutations. (**A**) COS-7 cells were used to express WT EGFP–VP40, EGFP–W191A, or EGFP–W191F and were imaged 12–14 h post-transfection. Each construct was imaged independently (3 times) and multiple cells were counted for each trial to determine the number of cells with pre-VLP structures emanating from the plasma membranes. White arrows indicate the presence of pre-VLPs in the respective images above. Insets are shown for both W191A and W191F to demonstrate differences observed in some cells for each mutant, where pre-VLPs were lacking. The W191A mutant also displayed perinuclear puncta in a large number of cells. Scale bars = 10 μm. (**B**) Quantification of cells imaged in (A) were plotted as normalized cells with VLPs (normalized to WT) from the three independent measurements. (**C**) VLP formation was assessed ~24 h post-transfection as previously described [16]. Precision Plus Protein^TM^ Kaleidoscope Pre-stained Protein Standards (BioRad, Hercules, CA, USA) were used and shown in the right most lane. The bands shown from top to bottom are 100 kDa, 75 kDa, 50 kDa, and 37 kDa. eVP40 was quantified by Western blot for eVP40 in purified VLPs or eVP40 in cell lysates (CL). GAPDH was used as a loading control. The 37 kDa band of the Precision Plus Protein^TM^ Kaleidoscope Pre-stained Protein Standards is shown in the right most lane. (**D**) Budding efficiency was normalized to WT and plotted for W191F and W191A. Statistical analysis was performed for panels (B) and (D) where a * indicates *p* < 0.05.

**Figure 3 pathogens-09-00402-f003:**
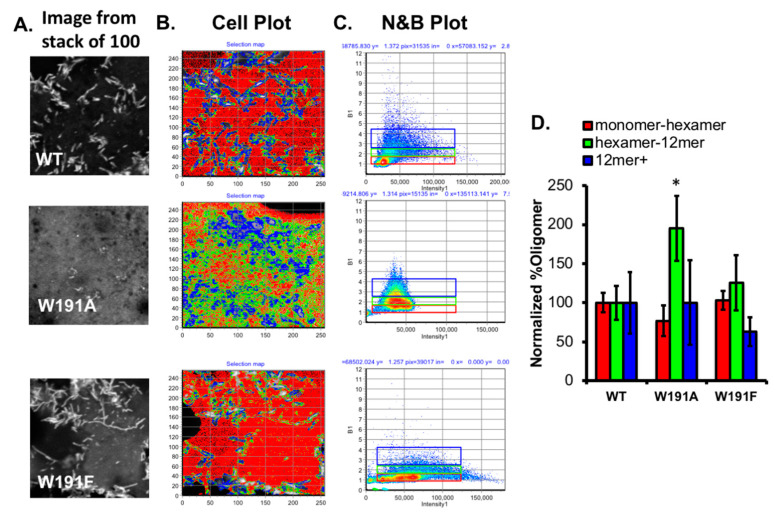
Oligomerization of VP40 and Trp^191^ mutations in COS-7 cells. (**A**) Raster image correlation spectroscopy (RICS) was performed on COS-7 cells expressing WT EGFP–VP40, EGFP–W191A, or EGFP–W191F. One image from a stack of 100 is shown to reflect the localization of VP40 throughout the image. (**B**) SimFCS was used to analyze the RICS data and determine the apparent oligomerization state of VP40 per pixel in the image. Red pixels indicate monomer–hexamers, green pixels indicate hexamer–12mer populations, and blue pixels indicate 12mer and greater populations. The color and oligomer populations reflect that shown in panels (C) and (D). (**C**) Brightness (*y*-axis) versus intensity (*x*-axis) was plotted (corresponding to different colors in images (B) and (D)). The selected colored boxes reflect the region selected for monomer–hexamer (red), hexamer–12mer (green), and 12mer and greater (blue). (**D**) N&B analysis was used to determine the number of monomer–hexamer, hexamer–12mer, and 12mer and greater populations of EGFP–VP40 or mutations throughout the images. Each oligomer population was normalized to WT to assess the differences between mutant and WT populations. * *p* < 0.05.

**Figure 4 pathogens-09-00402-f004:**
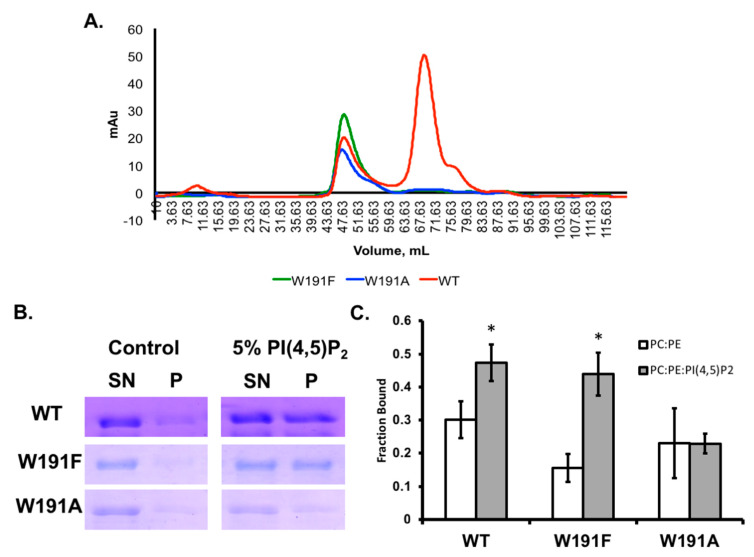
WT VP40 and Trp^191^ mutant dimer and octamer propensity and lipid binding analysis. (**A**) Size exclusion chromatography was performed post-affinity chromatography for WT VP40 (red), W191A (green), or W191F (blue). WT exhibited the characteristic dimer peak between 67 and 71 mL as well as an octamer peak at 47–51 mL. In contrast, W191A and W191F were predominantly octamers and dimer populations were barely detectable. The size exclusion column was routinely calibrated with high molecular weight standards from Cytiva (43,000–669,000 Da) (Marlborough, MA, USA). (**B**) A vesicle binding assay was performed as previously described [14] for WT, W191A, and W191F using control and PI(4,5)P_2_ containing vesicles where the bound (P—pellet) and unbound (SN—supernatant) fraction were separated by centrifugation. (**C**) Quantification of three independent experiments for the fraction of VP40 or mutant bound to control or PI(4,5)P_2_ containing vesicles. * *p* < 0.05.

**Figure 5 pathogens-09-00402-f005:**
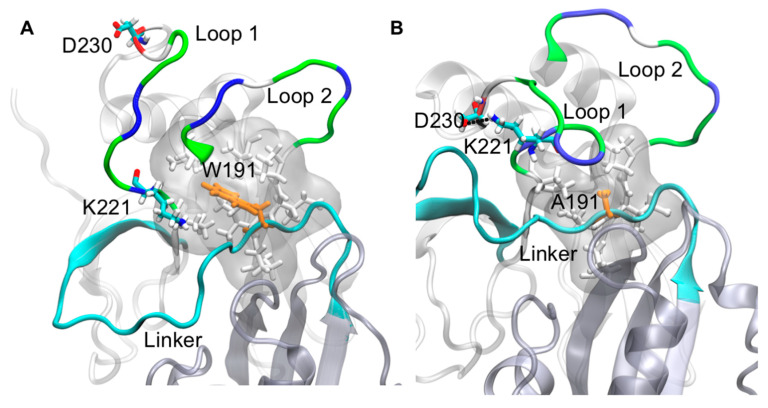
Molecular dynamics (MD) simulations of WT VP40 and W191A C-terminal domain flexibility. (**A**) MD simulations indicate that WT VP40 has a flexible lipid-binding Loop 1 (containing lipid-binding residues Lys^224^ and Lys^225^) and a fairly restricted lipid-binding Loop 2 (lipid binding residues Lys^274^ and Lys^275^) (See also Appendix A). (**B**) MD simulations with W191A demonstrate a significant change in Loop 1 flexibility due to a K221–D230 salt-bridge, a similar level of flexibility of Loop 2 (albeit a slightly different orientation), and a more flexible linker between the N-terminal domain (NTD) and CTD than that of WT.

**Figure 6 pathogens-09-00402-f006:**
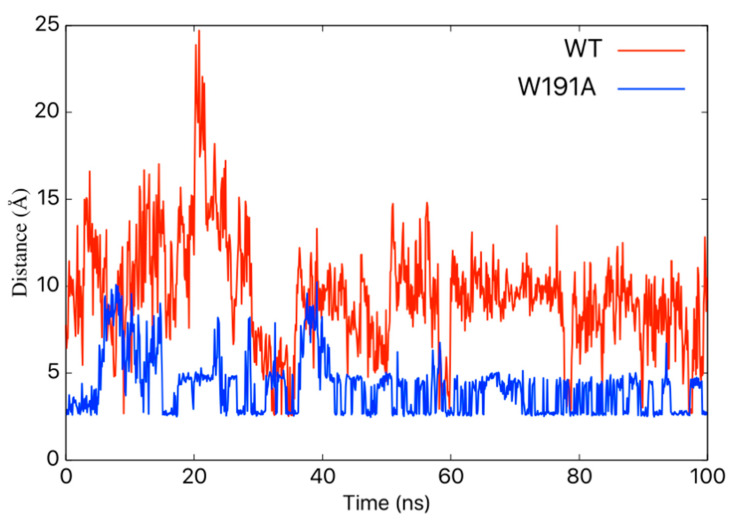
The hydrogen bond (N–O) distance between Lys^221^ and Asp^230^ side chains. MD simulations predicted changes in the distance between N–O in the Lys^221^ and Asp^230^ side chains for WT (red) and W191A (blue) VP40.

**Figure 7 pathogens-09-00402-f007:**
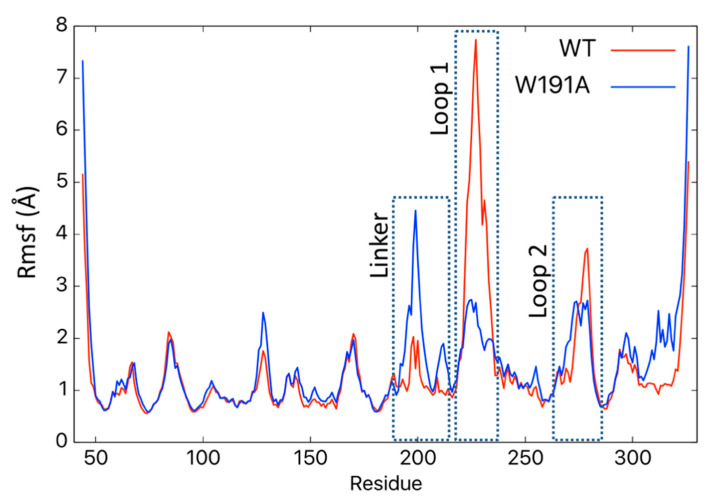
The root-mean-squared fluctuations (Rmsf) of VP40 residues in WT and W191A VP40. MD simulations predicted changes in the Rmsf in Å for different residues in WT (red) and W191A (blue) VP40. The regions for the NTD–CTD linker, Loop 1 and Loop 2 display the most significant changes in fluctuations between WT and W191A.

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
