# Peer review of "A Conserved Tryptophan in the Ebola Virus Matrix Protein C-Terminal Domain Is Required for Efficient Virus-Like Particle Formation"

_pathogens, 2020, doi:10.3390/pathogens9050402_

Round 1

Reviewer 1 Report

This paper details in depth and innumerable details the role of the eVP40 matrix protein in the formation of VLPs. Overall, the experiments are well planned and conducted, the results well described, the conclusions convincing and the review of the literature complete. I think it can be published as it is, the suggested changes are very minor.

Minor revision

Introduction: Although the modification of VP40 for Trp 95 (reviewed in the paper) and 191 are used to investigate the function-structure relationship, it could be interesting to cite if the inducted mutation polymorphisms at the same AA occur in nature by looking at the sequences deposited at the publicly accessible database.

Line 42 not limit to describe FDA approval and describe also WHO position on approved drugs and antibodies for disease treatment

Line 177 We then subjected….

Author Response

Introduction: Although the modification of VP40 for Trp 95 (reviewed in the paper) and 191 are used to investigate the function-structure relationship, it could be interesting to cite if the inducted mutation polymorphisms at the same AA occur in nature by looking at the sequences deposited at the publicly accessible database.

Response: Thank you for a great question.  We had reviewed those available and have not seen these mutations arise.

Line 42 not limit to describe FDA approval and describe also WHO position on approved drugs and antibodies for disease treatment

Response:  Excellent point.  We have made the revision.

Line 177 We then subjected….

Response:  Thank you.  The correction has been made.

Reviewer 2 Report

This is a well written and focused manuscript by an established group that explores the contribution of VP40 Trp191 to the ebolavirus lifecycle.  Two salient site directed mutants were made and studied in virus like particle (VLP) budding and fluorescence microscopy assays, to show the mutants were compromised to varying degrees in assuming normal plasma membrane locales and forming VLPs.  In vitro assays revealed the mutant proteins were biased in forming octamers over the wild-type dimer/ octamer mix and, together with the cellular data, indicates the fluidity of dimer to octamer transition is critical for efficient localization and virus egress.  The work is an important piece in the jigsaw puzzle of elucidating the precise mechanisms involved in choreographing matrix protein during ebolavirus assembly

L38 “…at an outbreak…” perhaps change to “….following an outbreak of hemorrhagic fever….”

Legend for figure 2C, please insert “(CL)” after “cell lysates”.

L176 “E. Coli” > “E. coli

Figure 4a, any comment on molecular weight standards to calibrate the column to be able to say one peak is dimer and the other is octamer?

L183 I’m not sure this sentence is quite summarizing or interpreting the cell data as one may see it.  Fig. 3D appears to show octamer present on a par with wt levels relative to the other forms, as expected, since VLP are produced (albeit at a lower level than wt) from the Trp>Phe mutation as shown in fig. 2c.  Since the Trp>Ala mutation appears totally compromised in VLP formation and is accumulating more VP40 in the cell lysate Western blot, isn’t the elevated level of octamer just a result of accumulation in this mutant rather than it being absent in the Phe system? 

Figure 5, is the resolution / prediction ever good enough to display hydrogen atoms?

L297 you may mean pET-46 rather than pet46.

L299-303 convention dictates adding 5’- and -3’ to oligonucleotides to eliminate ambiguity.

L311 please use symbol for micro instead of u.

L316 “anti mouse” > “anti-mouse”

[ L123 and L314 western is actually capitalized.  I didn’t think it was for decades, until a reviewer picked me up on it, but if you look back at the original literature, it is - ANALYTICAL BIOCHEMISTRY 112, 195-203 (1981).  Journals tend to vary about whether it is one way or another. ]

References 14, 22, 30, 40, 51, 54 – please check journal abbreviations, capitalizations, italicizations.

Author Response

L38 “…at an outbreak…” perhaps change to “….following an outbreak of hemorrhagic fever….”

Response:  Thank you.  The change has been made.

Legend for figure 2C, please insert “(CL)” after “cell lysates”.

Response:  Thank you.  The change has been made.

L176 “E. Coli” > “E. coli

Response:  Thank you.  The change has been made.

Figure 4a, any comment on molecular weight standards to calibrate the column to be able to say one peak is dimer and the other is octamer?

Response:  Thank you for this important comment.  We have inserted information in the figure legend on the molecular weight standards.

L183 I’m not sure this sentence is quite summarizing or interpreting the cell data as one may see it.  Fig. 3D appears to show octamer present on a par with wt levels relative to the other forms, as expected, since VLP are produced (albeit at a lower level than wt) from the Trp>Phe mutation as shown in fig. 2c.  Since the Trp>Ala mutation appears totally compromised in VLP formation and is accumulating more VP40 in the cell lysate Western blot, isn’t the elevated level of octamer just a result of accumulation in this mutant rather than it being absent in the Phe system? 

Response:  Thank you for this critical assessment.  The reviewer is most likely correct on this point and we have revised the text to include this assessment.

Figure 5, is the resolution / prediction ever good enough to display hydrogen atoms?

Response:  Thank you for this excellent question.  Since this is an all-atom simulation, the structure has the resolution to show the positions of hydrogen atoms. However, hydrogen atoms are generally quite flexible and assume different positions from one frame to next, except when hydrogen-bonded with other atoms. Thus, they are not generally shown in all-atom simulations due to these shortcomings of strength of predicted position.

L297 you may mean pET-46 rather than pet46.

Response:  Thank you.  The change has been made.

L299-303 convention dictates adding 5’- and -3’ to oligonucleotides to eliminate ambiguity.

Response:  Thank you.  The change has been made.

L311 please use symbol for micro instead of u.

Response:  Thank you.  The change has been made.

L316 “anti mouse” > “anti-mouse”

Response:  Thank you.  The change has been made.

[ L123 and L314 western is actually capitalized.  I didn’t think it was for decades, until a reviewer picked me up on it, but if you look back at the original literature, it is - ANALYTICAL BIOCHEMISTRY 112, 195-203 (1981).  Journals tend to vary about whether it is one way or another. ]

Response:  Thank you.  The change has been made.

References 14, 22, 30, 40, 51, 54 – please check journal abbreviations, capitalizations, italicizations.

Response:  Thank you.  The change has been made.

Reviewer 3 Report

Interesting and well explained study with clear results and figures. A statement on the broader significance of the finding might be beneficial. 

Author Response

Interesting and well explained study with clear results and figures. A statement on the broader significance of the finding might be beneficial. 

Response:  We thank the reviewer for the kind comments.  We have added statements to the broader significance.